# Advances in Imaging-Based Biomarkers in Renal Cell Carcinoma: A Critical Analysis of the Current Literature

**DOI:** 10.3390/cancers15020354

**Published:** 2023-01-05

**Authors:** Lina Posada Calderon, Lennert Eismann, Stephen W. Reese, Ed Reznik, Abraham Ari Hakimi

**Affiliations:** 1Urology Service, Department of Surgery, Memorial Sloan Kettering Cancer Center, New York, NY 10065, USA; 2Computational Oncology, Department of Epidemiology & Biostatistics, Memorial Sloan Kettering Cancer Center, New York, NY 10065, USA

**Keywords:** imaging, renal cell carcinoma, biomarker, renal mass, radiomics, radiogenomics

## Abstract

**Simple Summary:**

Current imaging techniques do not reliably distinguish renal cell carcinoma from other renal diseases. This review summarizes recent advances in other imaging methods for the diagnosis and monitoring of potential kidney tumors. Magnetic resonance imaging (MRI), positron emission tomography (PET)/CT using various radiolabeled molecules to detect specific cancer-associated features, and computational extraction of data from CT images have all proven useful for various purposes, but more research is needed to verify their reliability.

**Abstract:**

Cross-sectional imaging is the standard diagnostic tool to determine underlying biology in renal masses, which is crucial for subsequent treatment. Currently, standard CT imaging is limited in its ability to differentiate benign from malignant disease. Therefore, various modalities have been investigated to identify imaging-based parameters to improve the noninvasive diagnosis of renal masses and renal cell carcinoma (RCC) subtypes. MRI was reported to predict grading of RCC and to identify RCC subtypes, and has been shown in a small cohort to predict the response to targeted therapy. Dynamic imaging is promising for the staging and diagnosis of RCC. PET/CT radiotracers, such as ^18^F-fluorodeoxyglucose (FDG), ^124^I-cG250, radiolabeled prostate-specific membrane antigen (PSMA), and ^11^C-acetate, have been reported to improve the identification of histology, grading, detection of metastasis, and assessment of response to systemic therapy, and to predict oncological outcomes. Moreover, ^99^Tc-sestamibi and SPECT scans have shown promising results in distinguishing low-grade RCC from benign lesions. Radiomics has been used to further characterize renal masses based on semantic and textural analyses. In preliminary studies, integrated machine learning algorithms using radiomics proved to be more accurate in distinguishing benign from malignant renal masses compared to radiologists’ interpretations. Radiomics and radiogenomics are used to complement risk classification models to predict oncological outcomes. Imaging-based biomarkers hold strong potential in RCC, but require standardization and external validation before integration into clinical routines.

## 1. Introduction

Renal cell carcinoma (RCC) has an incidence of 12 per 100,000 in North America, and a peak incidence at the age of 60–70 years [1]. RCC incidence continues to rise, with an estimated 79,000 new cases and 13,920 deaths from RCC in 2022 in the United States alone [2]. The most common histologic subtype of renal cell carcinoma is clear-cell renal cell carcinoma (ccRCC), with five-year survival rates declining by stage of the disease. In the industrial world, the incidence of localized RCC continues to rise, with almost 70% of tumors being detected incidentally [3,4], secondary to increased utilization of abdominal imaging.

Current challenges in the treatment of renal cell carcinoma include diagnostic uncertainty, which leads to both under- and overtreatment of the disease. Conventional cross-sectional imaging techniques do not allow for the discrimination of malignant tumor sub-types, nor can they differentiate between benign lesions. Additionally, current imaging gives us opaque insight into patients with metastatic disease and its response to and progression with therapeutics. In recent years, research has focused on improved imaging techniques to enhance diagnostic precision and prognosis in patients with renal tumors. Advancements in the field have been multi-factorial, from enhancements in current ultrasound and cross-sectional imaging technologies, nuclear medicine studies, and the field of radiomics, which infers renal mass insights from radiologic data. In this review, we summarize and analyze the future of imaging modalities, and the advancements in radiomics and radiogenomics as they develop new ways of diagnosing and distinguishing renal cell masses (Table 1).

## 2. Magnetic Resonance Imaging (MRI)

Multiparametric MRI (mpMRI) allows for the evaluation of anatomic as well as functional characteristics of renal masses [5]. Specifically, diffusion MRI and perfusion MRI have been studied as imaging tools to aid in differentiating tumor histology or subtype, and assessing the response to treatment [6]. MRI has been proposed as an alternative to computed tomography (CT), which is limited in its ability to identify benign lesions such as fat-poor angiomyolipomas (AMLs) and oncocytomas [7,8]. Diffusion-weighted imaging (DWI) quantifies the mobility of protons that are associated with water (Brownian motion). Tissue that is highly cellular, such as tumors, restricts water molecules’ movement, and thus appears as a high-intensity signal on DWI, and has a low apparent diffusion coefficient (ADC) [9]. A systematic review and meta-analysis including four studies that used DWI to differentiate between malignant and non-malignant lesions showed DWI to have 86% sensitivity and 78% specificity. In this meta-analysis, DWI used to differentiate high-grade and low-grade RCCs had an area under the curve (AUC) of 83%, reflecting moderately accurate test performance. However, there were no standardized criteria to compare radiological findings to different imaging modalities or pathological specimens [10]. These values are comparable to those associated with CT scans, where the sensitivity has been reported to be 88% and the specificity 75% [11]. Using diffusion MRI, parenchymal wash index, and ADC ratio were correlated with clear-cell RCC Fuhrman grade, with a pooled sensitivity and specificity of DWI to differentiate between high and low grades of 78% and 86%, respectively [12,13].

Perfusion MRI, which assesses tissue perfusion at the micropapillary level, offers the possibility of improving performance characteristics. There are three main types of perfusion MRI: dynamic contrast-enhanced (DCE), dynamic susceptibility contrast (DSC), and arterial spin labeling (ASL) [6,14]. DCE and DSC calculate changes in signal intensity before and after intravenous gadolinium contrast injection, which measures perfusion parameters. ASL does not require intravenous contrast, and measures perfusion by detecting water protons in the blood [14]. Using ASL perfusion MRI, Lanzman et al. compared pre-operative MRI perfusions of 42 patients with various types of renal masses. The RCC histology was associated with different mean perfusion levels, with papillary RCC having lower perfusion levels than all other RCC types, and oncocytomas having significantly higher perfusion levels than RCCs [15]. 

Differentiating fat-poor AML and RCC based solely on imaging is a known challenge. In a systematic review and meta-analysis by Wilson et al., MRI was found to be 83% sensitive and 90% specific for the detection of fat-poor AMLs, with an AUC of 0.93 [16]. In a retrospective study of 109 renal masses, Kay et al. developed an MRI diagnostic algorithm comprising 11 MR imaging features to determine the most likely histology of a renal mass. They found a sensitivity and specificity of 85% and 76%, respectively, for predicting clear cell histology, and 80% and 94%, respectively, for predicting papillary histology. Their algorithm, however, was weak in predicting chromophobe, oncocytoma, and fat-poor AML histologies [17]. Using this algorithm, Canvasser and colleagues developed a clear cell likelihood scale of 1 (less likely) to 5 (most likely), and found a sensitivity of 78% and a specificity of 90% for scores of 4 and 5 [18]. The clear cell likelihood scale was evaluated in a larger retrospective cohort of 454 renal masses, and the authors found a 93% positive predictive value for a score of 5, and a sensitivity and specificity of predicting clear-cell RCC of 91% and 56%, respectively, for scores of 4 and 5 [19]. Although these scales do not provide insight into tumor aggressiveness, they may be used to help select treatment for small renal masses, and to determine candidates for surveillance [5]. 

There is a growing interest in using mpMRI, not only to predict renal mass histology and behavior, but also to assess response to therapy. In a prospective mixed cohort of treatment-naïve and exposed patients, Tsai et al. evaluated changes in tumor ASL MRI perfusion as a measure of response to sunitinib and pazopanib treatment for metastatic RCC. Perfusion on MRI imaging, as evaluated by objective response rate, was compared among 6 responders and 11 non-responders at multiple time points during treatment and up to disease progression. Responders had a higher baseline tumor perfusion than non-responders (404 mL/100 g/min vs. 199 mL/100 g/min; *p* = 0.02), suggesting this could aid in identifying responders to therapy with tyrosine kinase inhibitors [20]. In a prospective, randomized, double-blinded trial that compared sorafenib and placebo, DCE MRI was also evaluated as a pharmacodynamic biomarker of response to sorafenib in metastatic RCC. Of the 44 patients with two available MRIs for comparison, two DCE parameters (area under the contrast concentration versus time curve 90 s after contrast injection [IAUC90], and volume transfer constant of contrast agent [Ktrans]) were evaluated. Although patients with high baseline Ktrans had better progression-free survival (PFS) compared to patients with low baseline Ktrans (log-rank *p* = 0.027), there was no significant association between change in IAUC90 and Ktrans with PFS [21]. 

In summary, while multiple studies have evaluated the use of MRI to predict the histology and grade of renal masses, and to assess response to treatment in metastatic RCC, they were generally small studies that used a variety of non-standardized mpMRI metrics [14]. Future studies are needed to validate the use of these metrics and demonstrate their usefulness in clinical scenarios. 

## 3. Contrast-Enhanced Ultrasound

Ultrasound is a widely used diagnostic tool, and in many settings is the first modality used to evaluate renal pathologies. Focal lesions, hydronephrosis, and vascular pathologies can be identified, whereas benign lesions cannot be reliably distinguished from malignancies by conventional ultrasound [22]. Therefore, contrast-enhanced ultrasound (CEUS) has been proposed to visualize RCC characteristics. The contrast agent used for ultrasound is based on microbubbles, and amplifies the signal of microvascular structures [23]. CEUS has been shown to highly differentiate RCC from oncocytoma and angiomyolipoma [24,25]. In one study, combining CEUS parameters showed a 93% sensitivity and 100% specificity for renal malignancies [26]. Furthermore, a study of 85 patients with 93 renal masses showed that peak intensity and time to peak intensity in CEUS differed between clear-cell RCC, chromophobe RC and papillary RCC [27]. Additional studies have shown specific enhancement characteristics compared to clear-cell RCC [28]. CEUS has also been used in the diagnostic setting. For example, Lamuraglia et al., in 2006, showed that CEUS holds predictive value in metastatic RCC patients treated with the multi-kinase inhibitor sorafenib [29]. Similarly, Williams et al. reported significant changes in CEUS along with anti-angiogenic therapy of metastatic RCC, although CEUS parameters did not correlate with progression-free survival or best response rate to therapy [30]. Current research focuses on the assessment of CEUS to predict the response to immunotherapy in metastatic RCC (NCT05206942). 

In summary, CEUS is a diagnostic modality that offers potential advantages in the characterization of renal masses, including enhanced diagnostic performance, characterization of renal mass histologic subtypes, its low cost, its low barrier to access, and the absence of ionizing radiation. Additional studies are needed in larger settings to validate these findings, including understanding performance characteristics in patients with different habitus. 

## 4. Positron Emission Tomography–Computed Tomography (PET/CT)

Molecular or nuclear imaging studies rely on in vivo visualizations of biological processes at a cellular and molecular level, using radiopharmaceutical compounds that bind to a molecule of interest [5]. In RCC specifically, nuclear imaging allows for the identification of not only anatomic locations, but also for molecular pathways and processes that are associated with specific histologic features and tumor behavior. Multiple positron emission tomography (PET) radiotracers have been developed and studied as both prognostic and predictive biomarkers in RCC [6,31]. 

### 4.1. ^18^F-Fluorodeoxy-Glucose (FDG) PET/CT

While ^18^F-fluorodeoxy-glucose (FDG)-PET is the most common and well-known radiotracer used in other cancers, it has limited applicability in RCC due to its variable activity in primary and metastatic tumors, as well as physiologic uptake in normal renal parenchyma [5]. In a meta-analysis of 14 studies that assessed this modality in advanced RCC, the pooled sensitivity and specificity of FDG-PET/CT were 62% and 88%, respectively, for renal lesions, and 79% and 90%, respectively, for extrarenal lesions [32]. Despite variable uptake at the individual lesion level, the maximum standardized uptake value (SUVmax) of lesions in patients with advanced RCC has been independently associated with overall survival (OS) and PFS [33,34]. FDG-PET/CT activity has been proposed as a surrogate for tumor aggressiveness, as it has also been correlated with higher Fuhrman grade, TNM stage, and sarcomatoid features, and can aid in the prediction of progression and in clinical decision making [35,36,37,38]. Additionally, the detection of metastatic or recurrent sites was evaluated in a recent meta-analysis that included 14 studies [39]. The pooled sensitivity was described with 0.86, and specificity with 0.88 [39]. Accordingly, FDG-PET may be a useful re-staging tool for RCC, but current evidence is mostly based on retrospective studies, and lacks prospective investigations [39]. Hou et al. focused on the clinical value of FDG-PET in papillary RCC, and reported a similar sensitivity of 81% in the primary lesion and 100% in recurrent lesions [40]. These preliminary retrospective studies are promising, and need to be confirmed in larger prospective studies. Limitations to the use of FDG-PET/CT include practicality, cost, and variable results across multiple studies [41].

### 4.2. ^124^I-cG250 (^124^I-Girentuximab)

Girentuximab, formerly known as antibody cG250, is one of the most promising nuclear imaging methodologies in the characterization of solid renal masses [5]. It selectively binds to carbonic anhydrase IX (CA-IX), a protein that is overexpressed in VHL-mutated pathways in response to hypoxic conditions, and is expressed in 95 to 100% of clear-cell RCCs [6,42,43]. A multi-center phase III trial, the REDECT trial, evaluated the diagnostic efficacy of ^124^I-girentuximab PET/CT and contrast-enhanced CT (CECT) in identifying clear-cell RCC in patients with indeterminate renal masses that were scheduled for surgical resection. Imaging was performed 2–6 days after intravenous administration of girentuximab, and prior to surgical resection. Imaging readings were classified as clear-cell RCC and non-clear-cell RCC, which were then compared to final surgical pathologies. In 195 patients that had imaging and pathology available for analysis, the average sensitivity and specificity were 86.2% and 85.9%, respectively, for girentuximab-PET/CT, and 75.5% and 46.8%, respectively, for CECT. Furthermore, the inter-reader agreement was higher for girentuximab PET/CT [44]. Although the limitations of this study included a bias in patient selection, using only pre-surgical candidates, nevertheless, it provides the most accurate validation of pathology with imaging using PET/CT. Different radiotracers targeting CA-IX are currently being studied to improve clinical practice to reduce the long half-life of girentuximab. The molecule F-VM4-037, which is reported to have an 18-minute plasma half-life, has been studied in a phase II trial to allow same-day imaging [45]. Although the performance characteristics of this approach appear to be promising, logistics and timing remain ongoing barriers and, additionally, advancements in the technology will need to be validated in order for it to be used in clinical practice. 

There is currently a prospective, open-label, multi-center phase III trial evaluating the performance characteristics of girentuximab (an anti-CAIX monoclonal antibody) labelled with 89Zr, to evaluate indeterminate renal masses to differentiate clear-cell RCC from other renal masses (ZIRCON Trial; NCT03849118). Its preliminary results were reported recently, and exceed the predetermined sensitivity and specificity study targets, with the imaging agent delivering 86% sensitivity and 87% specificity [46]. The phase I study showed in all ten cases a good toxicity profile, and was able to differentiate between clear-cell RCC and non-clear-cell RCC renal mass [47]. This technology is also being examined for diagnostic and therapeutic purposes in the STARLITE 2 Phase II study which evaluates the efficacy of Lu177 conjugated to girentuximab + nivolumab (anti-PD-1) systemic therapy. In a theranostic approach, Girentuximab could be labelled with 177Lu, a beta- emitter, that could induce single-strand DNA breaks into RCC cells. These agents are promising, and future research will focus on their incorporation into clinical practice. 

### 4.3. Prostate-Specific Membrane Antigen (PSMA)–Targeted PET/CT

Prostate-specific membrane antigen (PSMA) is a cell surface protein that is overexpressed in prostate cancer, as well as in the neovasculature of some solid tumors, including RCC. PSMA-targeted imaging was first described in metastatic RCC by Demirci et al. in 2014 [48]. Small studies have reported the sensitivity of F-DCFPyL PSMA PET/CT in detecting distant metastases to range from 88.9% to 94.7%, compared with 66.7% to 78.0% for conventional CT scans [49,50,51]. For localized renal masses, Golan et al. found that the mean SUV_max_ of ^68^Ga-PSMA-11 PET/CT was significantly higher in malignant as compared to benign lesions, and its washout coefficient *K_2_* was significantly lower in cancerous tissue [52]. Gao et al. reported that SUVmax of the same tracer could effectively differentiate high vs. low (WHO/SIUP grade I-II vs. III-IV) grade in 36 cases of clear-cell RCC. Furthermore, ^68^Ga-PSMA-11 PET/CT could predict the presence of adverse histopathological characteristics, such as necrosis and sarcomatoid and rhabdoid features, with an AUC of 0.89 [53]. Both of these studies, however, have small sample sizes, and are not consistent with prior studies that show a high-background signal, limiting the evaluation of primary masses [54,55]. In general, most studies of PSMA-targeted PET/CT have included mostly clear-cell RCC, but the few non-clear-cell RCC lesions evaluated by this approach have shown lower uptake than surrounding renal parenchyma [55]. In particular, a meta-analysis described that PSMA PET/CT may also be suitable for chromophobe RCC, due to its relevant PSMA expression [56]. In contrast, only 13.6% of papillary RCC demonstrate a PSMA expression and therefore, FDG PET is the preferred dynamic imaging modality [56,57]. Based on the inconsistency of PSMA uptake in non-clear-cell RCC, PSMA PET is not appropriate for staging RCC subtypes other than clear-cell and chromophobe RCCs [56,58]. 

The specificity of PSMA PET/CT to patients with clear-cell RCC may limit its routine clinical use in the localized setting. However, this technique could potentially become useful in patients with metastatic disease as a way to measure treatment response or disease progression. Additional studies are warranted to validate these findings. 

### 4.4. ^11^C-Acetate PET-CT

The radiotracer 11C-acetate is actively incorporated into tumor cells and integrated into cellular lipid structures, and may be helpful in distinguishing between malignant and benign lesions [59]. In the imaging of RCC, ^11^C-acetate has shown high uptake rates in clear-cell RCC, and even higher uptake rates in papillary RCC [60]. Additionally, in comparison to FDG-PET/CT, ^11^C-acetate-PET/CT is reported to have better sensitivity for detecting RCC [61]. ^11^C-acetate was evaluated as part of a dual-tracer technique with FDG PET/CT for the differentiation of AML from RCC; it was reported to have a sensitivity of 94% and a specificity of 98% [62]. In a case report, ^11^C-acetate was reported to predict early response to sunitinib in metastatic RCC. In summary, ^11^C-acetate is a promising radiotracer that may have the potential to be used to stage RCC, but evidence is based on small sample sizes. This tracer may also be relevant to the differentiation of AML from RCC as part of a complex dual imaging technique

## 5. Single Photon Emission-Computed Tomography (SPECT Scan)

### ^99^Tc-Sestamibi SPECT/CT

As previously mentioned, a key limitation of several imaging modalities is their limited ability to distinguish benign from aggressive RCC tumors, such as clear-cell RCC and oncocytoma [63,64]. ^99^Tc-sestamibi, a widely used nuclear imaging agent, offers the ability to differentiate these tumors based on their mitochondrial content. ^99^Tc-sestamibi is a lipophilic cationic mitochondrial imaging agent that accumulates in cells with high mitochondrial content and low multidrug resistance (MDR) pump expression, which are characteristic of renal oncocytomas [5,65]. In contrast, clear-cell and chromophobe RCC masses have a higher MDR pump expression and low mitochondrial activity, although chRCC has generally higher mitochondrial activity than clear-cell RCC [65,66]. In a prospective study by Gorin et al., the use of preoperative ^99^Tc-sestamibi single photon emission computed tomography (SPECT)/CT in detecting oncocytomas was assessed in 50 presurgical patients with T1 renal masses, and results were compared to final surgical pathologies. The authors found a sensitivity of 87.5% and a specificity of 95.2% in differentiating oncocytomas from hybrid oncocytic/chromophobe tumors [67]. Sistani et al. validated these findings and found that in 29 patients with 31 renal masses, all oncocytic lesions were positive on ^99^Tc-sestamibi SPECT/CT, whereas uptake was low in chromophobe RCC and absent in other RCC subtypes [68]. In a 90-patient study by Asi et al., strong ^99^Tc-sestamibi uptake was observed in 10 of 10 oncocytomas, while none was seen in most malignant lesions, except in 5 chromophobe RCC and 3 oncocytic papillary RCC masses. Other benign pathologies, such as chronic sclerosis, fibroma, hydatid cyst, and angiomyolipoma, also showed no uptake. The authors reported a positive predictive value of 60% and a negative predictive value of 91.3% in predicting benign pathologies. Additionally, they reported a relative uptake of 0.49 as an optimal cutoff to discriminate oncocytomas from other pathologies [69]. Given the already widespread use of ^99^Tc-sestamibi SPECT/CT and the high concordance of imaging findings with pathologies, results support further evaluation of its use in the identification of oncocytomas and other benign renal lesions. 

## 6. Radiomic and Radiogenomic Biomarkers

### 6.1. Radiomics

Radiomics and radiogenomics are two closely related fields with promising developments in characterizing cancers, predicting their behavior, and assessing treatment response. Radiomics consists of high-throughput extraction of quantitative data and the application of high-order statistical models to medical imaging to yield more objective and detailed analyses [5,70]. To make results more reproducible and interpretations more reliable, radiomic features quantifying morphological, intensity related, textural, and co-occurrence characteristics of CT, FDG-PET, and T1-weighted MRI features are being standardized [70]. These predefined quantitative radiomic features can be integrated into algorithms and artificial intelligence models to predict malignancy, tumor histology, tumor grade, and molecular characteristics [5].

Several radiomics models, most using texture analysis, have been designed and reported to be highly accurate in differentiating benign from malignant renal masses [71,72,73]. Varghese et al. evaluated 31 texture metrics of contrast-enhanced CT on 174 renal masses, and compared these findings with surgical pathologies. They found that six specific texture analysis features—entropy, entropy of fast-Fourier transform magnitude, mean, uniformity, information measure of correlation 2, and sum of averages—had high AUC values, with a mean AUC of 0.87 for differentiating benign versus malignant renal masses [72]. Furthermore, Uhlig et al. used machine learning algorithms to predict the malignancy of renal masses using 120 standardized radiomic features, and their diagnostic accuracies (base on surgical pathology) were compared with the those of blinded radiologists’ assessments. The sensitivity and specificity of their models were higher than radiologists’ diagnoses (0.88 vs. 0.80; *p* = 0.045, and 0.67 vs. 0.50; *p* = 0.083, respectively), with an AUC of 0.83 compared to 0.68 (*p* = 0.47) [73]. 

Other studies have also used convolutional neural network (CNN), a type of deep learning algorithm that processes images using pixel data recognition, in order to create models to differentiate malignant masses based on imaging [74,75,76,77]. In one of the largest series, Xi et al. developed a CNN model that included clinical and radiologic MRI data of 1162 renal lesions. This model was superior to radiological experts’ interpretations, with an accuracy of 0.70 vs. 0.60 (*p* = 0.053), sensitivity of 0.92 vs. 0.80 (*p* = 0.017), and specificity of 0.41 vs. 0.35 (*p* = 0.450), respectively. 

Using radiomics, various models have been reported to distinguish specific histologic features in renal masses. The accuracy of some models has exceeded conventional interpretation of radiologists, including those for CT to differentiate fat-poor AML from clear-cell RCC [78,79], oncocytomas from chromophobe RCC [80], and papillary type I from papillary type II RCC [81,82]. Similarly, such models have also been used to determine the presence of sarcomatoid features in clear-cell RCC, and to predict their nuclear grade [83,84,85]. 

Radiomics has also been investigated as a means of predicting responses to targeted therapy. In a retrospective analysis of 39 patients with 87 metastatic sites, Goh et al. compared the correlation of contrast-enhanced CT texture analysis parameters, at baseline and after two doses of tyrosine kinase inhibitors, with their progression with those of standard criteria. They reported texture analysis to be an independent factor associated with time to progression, supporting its potential to improve assessment and predict good response to therapy [86]. 

Although radiomics has promising utility for the diagnosis and assessment of renal masses and their response to therapy, its potential is limited by a lack of generalizability and clinical application. A recent systematic review and meta-analysis of 57 publications by Ursprung et al., of which 34 involved machine learning and artificial intelligence, reported that several similar characteristics have been investigated, but they have not been introduced into clinical practice because of a lack of external validation and reproducibility. This may be due to limited access to the codes and images that are used for the analysis and creation of the models [87]. 

One limitation to current radiomic analysis of renal masses is the fact that in most studies, the findings are compared to surgical pathologies, creating the possibility of selection bias. Additionally, some radiomics studies only analyzed single segments of the tumor, and therefore could not assess intratumoral heterogeneity, which may be an important factor in RCC [87]. This remains a promising field; however, prospective, randomized, and multicenter trials are required to accelerate its incorporation into clinical practice.

### 6.2. Radiogenomics

Radiogenomics is the integration of radiomics with genetic data and molecular signatures, based on the underlying principle that genetic alterations lead to distinct phenotypic imaging characteristics [5,6,88]. For clear-cell RCC specifically, the thorough analysis of genetic alterations with prognostic significance has led to an increased interest in the relationship between genomic signatures and imaging characteristics [88]. 

Karlo et al. analyzed the relationship of contrast-enhanced CT findings to genetic alterations in *VHL*, *PBRM1*, *SETD2*, *KDM5C*, and *BAP1* genes in 233 patients with clear-cell RCC. For *VHL*, they described a mutation frequency of 53.2% and imaging characteristics of well-defined tumor margins, nodular tumor enhancement, and gross appearance of intratumoral vascularity. *KDMC5* and *BAP1* had mutation frequencies of 6.9% and 6.0%, respectively, and were significantly associated with evidence of renal vein invasion. *PBRM1* mutations were observed in 28.8% of patients and, together with *VHL* mutations, were significantly more common among solid clear-cell RCC. *BAP1*, *KDMC5*, and *SETD2* mutations, with a frequency of 7.3%, were absent in multi-cystic clear-cell RCC [89]. In a similar study by Shinagare et al., *BAP1* mutations were significantly associated with ill-defined margins and the presence of calcifications, while *MUC4* mutations were associated with an exophytic growth pattern [90].

Radiogenomics has also been studied and integrated into models that can predict outcomes and response to treatment. Jamshidi et al. demonstrated how targeted, noninvasive, imaging-based surrogates of molecular assays (SOMA) can be constructed and used to determine outcomes in clear-cell RCC. They developed the Radiogenomic Risk Score (RRS), using a library of CT imaging features that have been correlated to genetic signatures shown to predict oncological outcomes. They followed 70 patients prospectively, classified RRS as high vs. low, and showed that the RSS predicts disease-specific survival, with a median survival of 40 months in patients with high RRS vs. 120 months in patients with low RRS (*p* = 0.00024) [91]. In another study, the ability of RRS to predict radiologic PFS was evaluated in patients with metastatic RCC undergoing presurgical treatment with bevacizumab in a phase II clinical trial. Patients with high RRSs on pretreatment CT scans had a median radiological PFS of 6 months vs. >25 months for patients with low RRSs (*p* = 0.005). Furthermore, overall survival differed significantly between the two cohorts: 25 months among high-RRS patients vs. >37 months among those with low RRSs (*p* = 0.03) [92]. These results must be interpreted carefully, as imaging characteristics may reflect tumor biology, and not necessarily the response to treatment. 

The limitations of radiomics also apply to radiogenomics. In addition, the known genetic alterations in RCC have a very low prevalence, which may limit the utility of radiogenomics. Furthermore, image characteristics, as they relate to genetic alterations, may not be consistent in all phases of imaging, and given the complexity of molecular pathways, heterogeneity within the tumor, and change over time, it is challenging to make direct correlations to specific imaging findings. This field will continue to expand with the advancement of knowledge regarding the relationships among genetic alterations, molecular pathways, and prognosis and response to treatment [5,88]. 

## 7. Future Directions

There are currently multiple trials focusing on the imaging of RCC and indeterminate renal masses. An early stage study is currently investigating hyperpolarized ^13^C pyruvate MRI to differentiate benign from malignant renal masses (NCT04687969). Due to the characteristic increased lactate production in malignant tissue, the conversion of hyperpolarized ^13^C pyruvate to lactate can be visualized with MRI, and makes this noninvasive pathway specific technique a promising approach in visualizing renal masses [93]. Another trial is investigating the use of a machine learning algorithm in patients undergoing PET/MRI using [18F]-DCFPyL, a PSMA ligand, in solid tumors including renal masses to assess tumor aggressiveness (NCT04687969). Aggressiveness assessed in imaging will be compared to the final histopathology in 50 patients, and patients will undergo up to three scans to evaluate longitudinal differences. With respect to treatment response, Mittlmeier et al. reported a pilot study that revealed a potential approach to predict early response to tyrosine-kinase inhibitors in metastatic RCC using 18F-PSMA PET/CT [94]. The theranostics approach, involving molecular imaging and a subsequent targeted therapy using the same radiotracer, represents an exciting modality in clear-cell renal cell carcinoma. The natural association of neovascularization in clear-cell RCC and PSMA expression may allow for targeted therapy using 177Lu-PSMA in highly aggressive RCC [57,95]. Furthermore, a combination of PSMA-targeted therapy and immunotherapy may also be a promising approach [57]. 

## 8. Conclusions

Various imaging platforms are currently being studied that offer significant promise to better inform the diagnosis and prognosis of patients with RCC. Although most technologies described here are not yet ready for routine clinical use, advances in imaging will soon help clinicians make better informed management decisions. 

## Figures and Tables

**Table 1 cancers-15-00354-t001:** Summary of imaging-based parameters.

Imaging Technique/Model	Description	Advantages	Disadvantages
MRI			
Multiparametric MRI	DWI uses water particle movement to identify tumor-like tissue, which has slower movement of water particles, and calculated an apparent diffusion coefficient (ADC)	Can be used to calculate likelihood of cancer vs. non cancer Can predict Fuhrman grade with a of 78% and 86% sensitivity and specificity, respectively.	Studies use a variety of non-standardized parameters for MRI that have not been validated in a larger population setting Poor-lipid AMLs remain a challenge to distinguish from chromophobe RCC and oncocytoma
Perfusion MRI (DCE, DSC, ASL)	Works by assessing perfusion at the micropapillary level, calculating changes in signal before and after contrast (DCE and DSC) or detecting water protons in blood (ASL)	Different histologic subtypes of RCC have different perfusion coefficients. ASL MRI can be used to predict response to treatment with sunitinib and pazopanib, with responders having higher baseline tumor perfusion	
PET-CT			
18F-FDG	FDG binds to metabolically active tissue, signaling cancer activity. From a meta-analysis, pooled sensitivity to detect renal lesions is 62% and specificity 88%.	Proposed surrogate for tumor aggressiveness, with maximum SUV of lesions in patients with advanced RCC is independently associated with overall survival, also related to higher Fuhrman grade, higher stage and sarcomatoid features.	Limited applicability in RCC due to physiologic uptake in renal parenchyma. Limited also by practicality, cost, and variable results across multiple studies.
Girentuximab, Xr-Girentuximab to CA-IX	CA-IX is a protein that is overexpressed in VHL-mutated pathways and expressed in 95–100% of ccRCC. Average sensitivity and specificity of 86.2% and 85.9%, respectively for identifying ccRCC.	Studies are validated with surgical pathology. Multiple ongoing studies for different molecules that target CA-IX. Recently zirconium girentuximab showed promising sensitivity and specificity of 86% and 86% in identifying ccRCC.	Long half life time of girentuximab, where injection needs to be administered 2–6 days prior to imaging. Logistics and timing of molecule remain main barriers
Tc-MIBI SPECT/CT	^99^Tc-sestamibi accumulates in cells with high mitochondrial content and low multidrug resistance (MDR) pump expression, which are characteristic of renal oncocytoma. Sensitivity of 87.5% and a specificity of 95.2% in differentiating oncocytomas and HOCTs	Widespread and usability of ^99^Tc-sestamibi SPECT/CT and high concordance of imaging findings with pathology, results are promising in the identification of oncocytomas	Other benign pathologies such as chronic sclerosis, fibroma, hydatid cyst and angiomyolipoma don’t have any uptake.
PSMA/PET	PSMA is a cell surface protein that is expressed in prostatic tissue and also in neovasculature of some cancers, including RCC, specifically clear cell histology	Increased sensitivity of PSMA PET/CT in detecting distant metastasis, with sensitivities of 89–95%, compared to 67–78% with conventional CT scan Can predict presence of adverse histopathological characteristics (necrosis, sarcomatoid an rhabdoid features)	Evaluation of primary lesions is limited, and studies have small sample sizes. Non ccRCC masses have a low PSMA uptake. Use may be limited to metastatic clear cell histology.
C-acetate PET	^11^C-acetate is actively incorporated into tumor cells and is integrated in cellular lipid structures. Has high uptake rates in papillary and ccRCC.	Better sensitivity rates than FDG PET. Using dual tracer c-acetate and FDG PET, AML was differentiated from RCC with sensitivity and specificity of 94% and 98%, respectively.	Evidence based on small sample size studies Differentiating AML and RCC would require dual complex imaging techniques and there is no added information on histology.
Radiomics	Objective and detailed analysis of imaging characteristics analyzed via quantitative methods and statistical models. Specific morphological characteristics, texture analysis and intensity of different parameters within the tumor can be standardized and integrated into algorithms and AI models to predict tumor malignancy, histology, grade and molecular characteristics. Convolutional Neural network (CNN) is a deep learning algorithm that processes pixel and clinical data to create models to predict malignancy of renal masses.	Reported AUC of 0.87 for differentiating benign versus malignant renal masses Radiomic models have reported to be superior to conventional radiological interpretation of images in distinguishing histologic subtypes and presence of sarcomatoid features Investigated as a biomarker for response to therapy, and texture analysis was found to be an independent factor associated with time to progression in patients with metastatic RCC being treated with TKI	Lack of generalizability and clinical application. There is few external validity and reproducibility of studies because of insufficient access to cades and images that serve for the creation of the models. Most studies are compared to surgical specimens, which implies a selection bias. Intra-tumoral heterogeneity may not be accounted for as there are only few areas of the tumor that are used for imaging analysis and creation of models
Radiogenomics	Genetic pathways express with different phenotypic imaging characteristics, so radiogenomic is the integration of radiomics with genetic tumoral data and molecular signatures.	Reported genetic associations with imaging characteristics for *VHL*, *KDMC5*, *BAP1*, and *MUC4*. Radiogenomic Risk Score (RSS) was developed to identify CT imaging features that are correlated to genetic signatures that have shown to predict oncological outcomes. This risk score was shown to correlate with progression free survival and response to treatment	Known genetic alterations in RCC have a very low prevalence, and are mostly applicable to ccRCC Given the complexity of molecular pathways, heterogeneity within the tumor and change in time, it is challenging to make direct correlations of gene and molecular pathways to specific imaging findings

MRI: magnetic resonance imaging; CT: computerized tomography; DWI: Diffusion-weighted imaging; DCE: dynamic contrast-enhanced; DSC: dynamic susceptibility contrast; ASL: arterial spin labeling; RCC: renal cell cancer; FDG: fluorodeoxy-glucose; SUV: standardized uptake value; CA-IX: carbonic-anhydrase XI; ccRCC: clear cell RCC; HOCTs: hybrid oncocytic/chromophobe tumors; AI: artificial intelligence; AUC: area under the curve; TKI: Tyrosin kinase inhibitors.

## Data Availability

Not applicable.

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
