# Peer review of "Advances in Imaging-Based Biomarkers in Renal Cell Carcinoma: A Critical Analysis of the Current Literature"

_cancers, 2023, doi:10.3390/cancers15020354_

Round 1

Reviewer 1 Report

The authors propose a very interesting narrative review of renal cell carcinoma imaging, an unmet need for daily clinical practice. However, I found a few inconsistencies that should be fixed before publication. I suggest the following major revisions:

1)    The simple summary should be revised as it is never reported which kind of tumor the authors talk about

2)    99Tc-sestamibi is not a PET probe and should be separated by PET tracers and discussed in a proper section, as it is misleading as reported by the authors. This should be clarified also in the abstract

3)    PSMA-ligands PET is for sure the most promising nuclear medicine technique in RCC, due to the very interesting preliminary results in extra-renal disease evaluation and to the wide availability that this technique will have in the near future. However this is not clear from the authors literature review. I suggest the authors to read and include a recent review on PSMA PET in RCC (https://doi.org/10.1007/s00432-022-03958-7) that could help them to better discuss PSMA PET in non clear cell RCC (see and include https://doi.org/10.1007/s11307-018-1271-2 ), in TKI response assessment (https://doi.org/10.1007/s00259-020-05165-3) and in a potential theranostic approach (PMID: 33897902). Moreover, 18)F]DCFPyL PET, described in future direction section, is a PSMA-ligand PET probe and should be moved into PSMA section.

A few minor revisions:

1)    Several typo errors that should be fixed along the text: line 11 “does”; line 97 “an 93%”; line 241 “SUVmax with PSMA PET” is incorrect; line 253 “a useful”; line 260 “my”

2)    In the introduction it is repeated “current imaging”. The authors should make explicit what they refer to (CT?) and a few words to describe it

3)     FDG PET paragraph. The authors should also include a more recent metanalysis than the one reported (doi: 10.1097/MNM.0000000000000618). Moreover they should add that RCC is not a typical Warburg tumor (see and include https://doi.org/10.3389/FONC.2016.00201) and reserve a few words for papillary RCC on FDG PET (see and include https://doi.org/10.1186/S40644-021-00393-8)

4)    Lines 203-204, “in theory” is unacceptable. Please rephrase in something like “In a theranostic approach Girentuximab could be labelled with 177Lu, a beta- emitter, that could induce single strand DNA breaks into RCC cells”

5)    Lines 209-210 “such as chromobpobe RCC and oncocytoma” why chromophobe? I would have taken clear cell RCC vs oncocytoma

6)    Line 236. PSMA is not overexpressed in normal prostatic tissue, but in prostate carcinoma cells’ surface. Please correct

7)    Lines 322-324. I don’t get why comparing radiomics to surgical pathology should be a bias

Author Response

Dear Editorial Committee,

Thank you for taking the time to review our manuscript: Advances in Imaging-based Biomarkers in Renal Cell Carcinoma: A Critical Analysis of the Current Literature.  Please see our responses to your recommendations. We think this changes have significantly improved the manuscript and we very much appreciate your feedback.

Sincerely,

Lennert Eismann, MD

Memorial Sloan Kettering Cancer Center

Reviewer 1:

The authors propose a very interesting narrative review of renal cell carcinoma imaging, an unmet need for daily clinical practice. However, I found a few inconsistencies that should be fixed before publication. I suggest the following major revisions:

1) The simple summary should be revised as it is never reported which kind of tumor the authors talk about

  • Thank you for this important remark. We revised the simple summary and point out focusing on renal cell carcinoma and kidney tumors. (See line 12-14)

2) 99Tc-sestamibi is not a PET probe and should be separated by PET tracers and discussed in a proper section, as it is misleading as reported by the authors. This should be clarified also in the abstract

  • Thank you for this valuable comment. We separated the 99Tc-sestamibi from the PET tracers and opened a new section for SPECT to avoid misunderstandings. (Now line 33-34 and 276-303)

3) PSMA-ligands PET is for sure the most promising nuclear medicine technique in RCC, due to the very interesting preliminary results in extra-renal disease evaluation and to the wide availability that this technique will have in the near future. However this is not clear from the authors literature review. I suggest the authors to read and include a recent review on PSMA PET in RCC (https://doi.org/10.1007/s00432-022-03958-7) that could help them to better discuss PSMA PET in non clear cell RCC (see and include https://doi.org/10.1007/s11307-018-1271-2 ), in TKI response assessment (https://doi.org/10.1007/s00259-020-05165-3) and in a potential theranostic approach (PMID: 33897902). Moreover, 18)F]DCFPyL PET, described in future direction section, is a PSMA-ligand PET probe and should be moved into PSMA section.

  • Thank you for those interesting suggestions. We added all the recommended literature to our manuscript (See line 232; 245-262 and new references 59-62)

A few minor revisions:

1) Several typo errors that should be fixed along the text: line 11 “does”; line 97 “an 93%”; line 241 “SUVmax withPSMA PET” is incorrect; line 253 “a useful”; line 260 “my”

  • Thank you very much for these corrections. We have fixed those typo errors and checked the whole manuscript again.

2) In the introduction it is repeated “current imaging”. The authors should make explicit what they refer to (CT?) and a few words to describe it

  • We appreciate this fine remark and re-wrote the section more explicit. (see line 59-60 and 65-66)

3) FDG PET paragraph. The authors should also include a more recent metanalysis than the one reported (doi: 10.1097/MNM.0000000000000618). Moreover they should add that RCC is not a typical Warburg tumor (see and include https://doi.org/10.3389/FONC.2016.00201) and reserve a few words for papillary RCC on FDG PET (see and include https://doi.org/10.1186/S40644-021-00393-8)

  • We added those interesting information and literature to our manuscript (see line 177-179; line 185-189)

4)    Lines 203-204, “in theory” is unacceptable. Please rephrase in something like “In a theranostic approach Girentuximab could be labelled with 177Lu, a beta- emitter, that could induce single strand DNA breaks into RCC cells”

  • Thank you for this remark. We rephrase the section as the reviewer suggested. (see line 227-229)

5)    Lines 209-210 “such as chromobpobe RCC and oncocytoma” why chromophobe? I would have taken clear cell RCC vs oncocytoma

  • We revised that sentence as the reviewer has suggested (see line 279).

6)    Line 236. PSMA is not overexpressed in normal prostatic tissue, but in prostate carcinoma cells’ surface. Please correct

  • Thank you for this important correction. (see line 232)

7)    Lines 322-324. I don’t get why comparing radiomics to surgical pathology should be a bias

  • We have deleted this point.

Reviewer 2 Report

Dear authors,

thank you for this interesting review. The manuscript is well-written and structured and the important approaches for future image analysis are described.

I only have minor comments:

- please describe what you mean with "current imaging techniques" (CT/MRI/ultrasound? what the guidelines recommend?); therefore, what are the current imaging findings for malignancy and what are limitations to highlight the clinical need for new develpements in the field.

- please provide a figure with a schematic overview of the approaches described in the manuscript and highlight the most important publications

Author Response

Dear Editorial Committee,

Thank you for taking the time to review our manuscript: Advances in Imaging-based Biomarkers in Renal Cell Carcinoma: A Critical Analysis of the Current Literature.  Please see our responses to your recommendations. We think this changes have significantly improved the manuscript and we very much appreciate your feedback.

Sincerely,

Lennert Eismann, MD

Memorial Sloan Kettering Cancer Center

Reviewer 2:

Dear authors,

thank you for this interesting review. The manuscript is well-written and structured and the important approaches for future image analysis are described. 

I only have minor comments:

1) Please describe what you mean with "current imaging techniques" (CT/MRI/ultrasound? what the guidelines recommend?); therefore, what are the current imaging findings for malignancy and what are limitations to highlight the clinical need for new develpements in the field.

  • We clarified the term “current imaging techniques” (see line 12; line 21-22). The clinical need for developing new diagnostic tools in renal/masses and RCC was highlighted again in line 24-25.

2) Please provide a figure with a schematic overview of the approaches described in the manuscript and highlight the most important publications

  • Thank you so much for this comment. We really struggled to think about how best we could represent the more salient point in the manuscript in a diagram, however have not found a good way to represent. We really think our primary table highlights the big take home messages for the topic and given the diversity of publications and narrative review format, that it would be hard to highlight some of these additional points.

Round 2

Reviewer 1 Report

The authors fixed some of my previous comments. However, FDG paragraph does not look to have been modified according to my suggestions. I still recommend to include a more recent metanalysis than the one reported (doi: 10.1097/MNM.0000000000000618) and to discuss papillary RCC (see and include https://doi.org/10.1186/S40644-021-00393-8).

Moreover, PSMA PET paragraph has been only partially modified and references have not been significantly changed. I still suggest to add 2 references at line 235 along with reference 54 (https://doi.org/10.1007/s00432-022-03958-7 and https://doi.org/10.1007/s11307-018-1271-2) as you can not affirm that sentence reporting only a 6 patients preliminary study as reference. Moreover please mention future perspectives, such as TKI response assessment (this work should be added: https://doi.org/10.1007/s00259-020-05165-3) and a potential theranostic approach (PMID: 33897902).

Lines 389-390 "[18F]-DCFPyL" is one of many abilable PSMA-ligands and should be clarified.

Author Response

Dear Editorial Committee,

Thank you again for taking the time to re-review our manuscript: Advances in Imaging-based Biomarkers in Renal Cell Carcinoma: A Critical Analysis of the Current Literature.  Please see our responses to your recommendations. We think these changes have again significantly improved the manuscript and we very much appreciate your feedback.

Sincerely,

Lennert Eismann, MD

Memorial Sloan Kettering Cancer Center

Reviewer 1:

The authors fixed some of my previous comments. However, FDG paragraph does not look to have been modified according to my suggestions. I still recommend to include a more recent metanalysis than the one reported (doi:10.1097/MNM.0000000000000618) and to discuss papillary RCC (see and include https://doi.org/10.1186/S40644-021-00393-8).

Thank you for those valuable suggestions. We fixed that paragraphe with the recommended literature.

  • The recent meta-analysis (doi:10.1097/MNM.0000000000000618) was added in line 182-186. New reference 39
  • The use of FDG-PET in papillary RCC (https://doi.org/10.1186/S40644-021-00393-8) was added in line 186-189. New reference 40

Moreover, PSMA PET paragraph has been only partially modified and references have not been significantly changed. I still suggest to add 2 references at line 235 along with reference 54 (https://doi.org/10.1007/s00432-022-03958-7 and https://doi.org/10.1007/s11307-018-1271-2) as you can not affirm that sentence reporting only a 6 patients preliminary study as reference.

We thank the reviewer for this excellent remark.

  • We added the Urso et al (https://doi.org/10.1007/s00432-022-03958-7 )in line 244-247. New reference 57
  • We added the Yin et al (https://doi.org/10.1007/s11307-018-1271-2) in line 247-249. New reference 59

Moreover please mention future perspectives, such as TKI response assessment (this work should be added: https://doi.org/10.1007/s00259-020-05165-3) and a potential theranostic approach (PMID: 33897902).

Thank you for this comment.

  • We added Mittlmeier et al. in line 408-410. New reference 95
  • We added Gorin et al. and Toyama et al. in line 410-415. New references 58 and 96

Lines 389-390 "[18F]-DCFPyL" is one of many abilable PSMA-ligands and should be clarified.

  • We clarified the PSMA ligand in line 405.

Round 3

Reviewer 1 Report

FDG and PSMA paragraphs have been improved and are now consistent with literature data. I think the manuscript is now ready to be accepted for publication.

When submitting proofs, please correct at line 223 "F-DCFPyL PSMA PET/CT". The correct name of the PET probe should be "18F-DCFPyL PET/CT, which is one of the many PSMA-ligands available)"